# Industrial-Grade Graphene Films as Distributed Temperature Sensors

**DOI:** 10.3390/s25103227

**Published:** 2025-05-21

**Authors:** Francesco Siconolfi, Gabriele Cavaliere, Sarah Sibilia, Francesco Cristiano, Gaspare Giovinco, Antonio Maffucci

**Affiliations:** 1Department of Electrical and Information Engineering, University of Cassino and Southern Lazio, 03043 Cassino, Italy; gcavaliere@unisa.it (G.C.); sarah.sibilia@unicas.it (S.S.); maffucci@unicas.it (A.M.); 2EUT+ Institute of Nanomaterials and Nanotechnologies-EUTINN, European University of Technology, European Union, 03043 Cassino, Italy; giovinco@unicas.it; 3Department of Information and Electrical Engineering and Applied Mathematics, University of Salerno, 84084 Fisciano, Italy; 4E-Lectra srl, 03043 Cassino, Italy; 5Nanesa srl, 00144 Roma, Italy; francesco.cristiano@nanesa.com; 6Department of Civil and Mechanical Engineering, University of Cassino and Southern Lazio, 03043 Cassino, Italy

**Keywords:** graphene nano-platelets, humidity sensors, nanomaterials, temperature sensors

## Abstract

This paper investigates the feasibility of a multi-purpose use of thin films of industrial-grade graphene, adopted initially to realize advanced coatings for thermal management or electromagnetic shielding. Indeed, it is demonstrated that such coatings can be conveniently used as distributed temperature sensors based on the sensitivity of their electrical resistance to temperature. The study is carried out by characterizing three nanomaterials differing in the percentage of graphene nanoplatelets in the temperature range from −40 °C to +60 °C. The paper demonstrates the presence of a reproducible and linear negative temperature coefficient behavior, with a temperature coefficient of the resistance of the order of $-1.5\cdot 10^{-3}$
${{}^{\circ}C}^{-1}$. A linear sensor model is then developed and validated through an uncertainty-based approach, yielding a temperature prediction uncertainty of approximately ±2 °C. Finally, the robustness of the sensor concerning moderate environmental variations is verified, as the errors introduced by relative humidity values in the range from 40% to 60% are included in the model’s uncertainty bounds. These results suggest the realistic possibility of adding temperature-sensing capabilities to these graphene coatings with minimal increase in complexity and cost.

## 1. Introduction

In recent decades, graphene-based materials have garnered increasing attention due to their exceptional mechanical, thermal, and electrical properties, which enable the development of innovative applications across various fields such as electronics [1], electromagnetics [2], materials science [3], chemistry and biology [4], medicine [5], and environmental monitoring [6], among others [7,8]. Following numerous successful demonstrations of proof-of-concept devices, recent research has shifted focus toward the industrial scalability of graphene technologies. In this context, industrial-grade graphene has emerged as a cost-effective alternative to single- or few-layer graphene, which remains expensive to produce at scale [9]. Composite materials that incorporate graphene nanoplatelets (GNPs), carbon nanotubes (CNTs), and hybrid fillers present a promising balance between performance, production volume, and economic sustainability [10,11]. A desirable feature of these materials is their multifunctionality, which allows for the simultaneous realization of electrical, mechanical, and thermal functionalities [12,13].

This work capitalizes on that potential, proposing commercial GNP-based composite coatings—initially designed for thermal management or electromagnetic shielding—as low-cost, distributed temperature sensors without requiring modifications to the existing material or its industrial fabrication process. Using GNPs to realize nanocomposite materials is one of the most popular options for obtaining industrial-grade graphene. This material can balance good physical properties [14,15,16,17] and industrial scalability. Indeed, GNPs can be efficiently fabricated by techniques such as wet-jet milling [18], liquid exfoliation [19], or microwave irradiation [20].

Several studies have investigated using graphene and related materials in temperature-sensing applications. High-performance sensors made from monolayer [21] or few-layer graphene [22] exhibit Temperature Coefficients of Resistance (TCR) ranging from −0.001 to −0.01 ${{}^{\circ}C}^{-1}$ achieving sub-degree precision and fast response times. Interdigitated structures based on graphene oxide have demonstrated improved sensitivity and potential for miniaturization [23]. Highly linear graphene-based temperature sensors have been recently demonstrated, with a linearity of 0.999 in the range of 30–100 °C, a sensitivity of 0.05% ${{}^{\circ}C}^{-1}$, quick response (2 s), and short recovery time (3.9 s) [24]. Fast temperature sensors working in a wide range (−190 °C, +280 °C) have been obtained using reduced graphene oxide films [25]. Additionally, hybrid composites that combine carbon nanotubes, graphene nanoplatelets, and polymers may enhance sensor performance while adding flexibility and integration capabilities [26]. For instance, a graphene-polydimethylsiloxane composite doped with polyaniline has been used to realize a highly linear (*R*^2^ = 0.999, where *R*^2^ is the coefficient of determination, i.e., the square of the sample correlation coefficient) and highly sensitive (1.60% ${{}^{\circ}C}^{-1}$) temperature sensor in the range 25–40 °C, with a fast response (0.7 s) [27]. Despite these promising results, in most cases, such sensors have not yet reached the market due to fabrication limits that make them not competitive compared to classical temperature sensors. Indeed, the mentioned responses can be achieved only by using high-quality graphene materials, which are usually produced by small-scale and expensive processes such as chemical vapor deposition (CVD). On the other hand, their multi-functional nature suggests using graphene-based materials to realize multi-sensing platforms such as those used for health monitoring systems [26,28]. For instance, in [28], laser-induced graphene and laser-reduced graphene are used to realize body temperature sensors, micro-supercapacitors, and electrodes for performing electrocardiograms.

Following this approach, this work investigates the possible dual-use of a graphene coating industrially made by GNPs mixed with polymeric binders as a distributed temperature sensor. Therefore, the main scope of this work is not proposing a novel and highly performing graphene-based temperature sensor but demonstrating the possibility of adding the functionality of temperature sensing to an industrial graphene coating primarily chosen for other purposes. The graphene material adopted for such coating is a composite made by a high percentage of GNPs using an industrial process developed by Nanesa srl, Roma, Italy [29].

The fabrication process and the structural characterization are described in Section 2. The operating principle is based on the sensitivity to the temperature of the electrical resistance of such materials. The main physical mechanisms are discussed in Section 2, where the resulting models for the resistance are also shown. Then, the sensor model is proposed, and the experimental setup for its characterization and validation is presented. The estimation of the sensor uncertainty is also discussed. In Section 3, the results of the experimental analysis, combined with the model predictions, allow us to obtain the key findings of this work: (i) the demonstration of the feasibility; (ii) the validation of a linear sensor model with quantified uncertainty; (iii) the quantification of the impact of humidity on the sensor’s performance.

## 2. Materials and Methods

### 2.1. GNP Strips Fabrication and Characterization

The industrial graphene composites analyzed in this paper are fabricated in strips 1 cm wide and 10 cm long with a thickness of 85–100 µm. These films are made using standardized formulations and developed using well-established processing methods and quality control procedures tailored by the manufacturer (Nanesa [29]).

Specifically, the industrial process starts from a graphitic precursor, from which graphene nanoplatelets (GNPs) are produced through thermal expansion and liquid-phase exfoliation [18,19]. Figure 1a presents a Scanning Electron Microscope (SEM) micrograph of a single GNP, taken with a ZEISS LEO 1530 FEG, 5 kV, 5 mm working distance (Oberkochen, Germany). These GNPs have typical lateral dimensions in the tens of microns and average thicknesses of approximately 12 nm. The GNP has an aspect ratio of 3.100, which indicates a highly exfoliated material. Other techniques suitable for mass production, such as wet-jet milling [18] and microwave irradiation [20], can achieve similar characteristics.

GNPs are then dispersed in acetone or an aqueous solution to be subjected to sonication; a polymeric binder is included during this phase. The choice of solvent depends on the desired formulation: acetone is preferred for its fast evaporation rate and better compatibility with hydrophobic polymer matrices, while aqueous dispersions are more suitable when using hydrophilic additives, such as boron nitride, and for environmentally friendly processing [30].

To produce GNP sheets, the liquid formulation is deposited onto a release substrate using a controlled spray system operated by a pantograph. This setup allows for automatic deposition with controlled coating weight. Key process parameters—air and liquid pressures, nozzle-to-substrate distance, movement speed, and number of passes—are preset based on the specific product. Typically, the pressure values range from 0.9 to 1.5 bar, the nozzle-to-substrate distance from 10 to 20 cm, the movement speed from 60 to 120 cm/min, and the number of deposition cycles from 1 to 8. Following the spray deposition, the coating is compacted via cold calendering, applying a uniform pressure of up to 10 kN/m. The calendering process compacts the strips and optimizes their thickness-to-alignment ratio, improving electrical conductivity and film homogeneity [31].

The SEM images provided in Figure 1b,c show the random structure of the GNP agglomerate and a side view of the GNP film, respectively. The properties of the resulting composite are heavily influenced by the size of the graphene nanoplatelets (GNPs) used in the manufacturing process. To achieve superior thermal and mechanical performance, thinner GNPs are preferable [32]. At the same time, for enhanced electrical conductivity, the size/thickness aspect ratio, size, and surface area [33] of the GNPs should be considered. Therefore, it is essential to use manufacturing processes that allow precise control over the dimensions of the GNPs. The quality of the manufacturing process in terms of defects is checked by a Raman spectroscopy analysis, whose result is plotted in Figure 2. Indeed, the low intensity and broadness of the D-band associated with the high intensity and sharpness of the G-band indicate a low concentration of defects in the surface of the material, assessing the degree of structural disorder, as shown in [34,35]

Table 1 summarizes the characteristics of the industrial graphene strips analyzed in this paper. The first two materials (G−PREG 95/5 and 70/30) are composites with a very high percentage of GNPs, where the binder is mainly intended to provide mechanical properties to the strip. The Authors have previously derived the electro-thermal parameters associated with these two materials in [36,37,38]. The third material is made by using a hybrid filler strategy, specifically by mixing GNPs and boron nitride (BN) at equal fractions (47.5%) together with a small percentage of polyurethane (5%). This formulation increases the strip’s electrical resistance without lowering the thermal conductivity, leveraging BN’s electrically insulating yet thermally conductive properties [30].

The reason for using 1 cm × 10 cm GNP strips is related to the need to investigate macroscopic samples with an aspect ratio that simplifies the measurement of the electrical resistance, allowing the use of a simple test fixture (see Section 2.3). Instead, the sheet thickness is only determined by the selection of standardized formulations provided by the manufacturer.

### 2.2. Operating Principles of the Sensing Element: Modeling the Impact of the Temperature on the Electrical Resistance

The present work’s operating principle depends on the temperature of the electrical resistance of the graphene strips. The theoretical justification of this principle is provided by the many models available in the literature, which describe the influence of temperature variations on the electrical transport of these materials. However, this dependence is studied for nanocomposites made by a small fraction of GNP reinforcement in many works. In particular, the behavior of the electrical resistance is investigated in composites where the GNP percentage is close to the percolation threshold (below and above it), which usually occurs with a few percent of GNPs in weight and/or volume [31,39,40]. In [31,40], the electrical resistance is derived from experimental characterizations. Hence, the observed influence of temperature is not given a physical interpretation. Instead, in works like [41] the relation between resistance and temperature is studied using a simple equivalent 3D resistive network analyzed using Monte Carlo simulations.

From a theoretical point of view, the mobility of charge carriers in carbon nanomaterials is affected by two competing mechanisms: thermal activation, which increases carrier mobility by enabling more carriers to conduct, and carrier scattering, which tends to reduce mobility due to phonon interactions. The first mechanism dominates semiconducting systems, leading to decreased resistance as temperature rises; the second dominates in metallic systems, where resistance increases with temperature [26,42].

In simple nanostructures such as individual carbon nanotubes (CNTs) or graphene nanoribbons (GNRs), and even for aligned CNT bundles or GNR arrays, the distributed resistance $R_{d}$ may be modeled as a function of temperature *T* as follows [42]:(1)$$R_{d}\left(T\right)=\frac{R_{0}}{l_{mfp}\left(T\right)M\left(T\right)}len$$
where *len* is the length of the structure, $R_{0}=12.9\;k\Omega$ is the quantum resistance, *M* is a quantum parameter named as the effective number of conducting channels, and $l_{mfp}$ denotes the mean free path. In this model, the thermal activation is associated with the number of conducting channels, which can increase with temperature (for multi-walled CNTs and GNRs) or remain almost constant (for single-walled CNTs) [43,44,45]. Instead, the scattering process is responsible for a mean free path decreasing with the temperature increase [46].

Therefore, Equation (1) predicts the possibility of having any sign for the derivative of the resistance with respect to the temperature, depending on the combined effect of these two competing mechanisms. When the thermal activation overcomes the scattering effects, these materials exhibit a Negative Temperature Coefficient (NTC) behavior, with an electrical resistance decreasing with the temperature [46]. For instance, in metallic single-walled CNTs, the number of conduction channels *M* is almost independent of temperature [44]. Thus, the thermal activation has no effect, and the resistance increases with temperature, dominated only by the scattering. Instead, when the temperature of multi-walled CNTs or GNRs increases, the thermal activation leads to a monotonically increasing effective number of conducting channels [44,45] that can counteract the decrease, resulting in an NTC behavior, as experimentally observed, for instance, in [46].

Simple analytical expressions like that in Equation (1) cannot describe the resistance of the material analyzed in this work since it is composed of irregular GNP flakes dispersed in a dielectric binder. However, a simplified interpretation of the charge transport can still provide insight into the experimentally observed NTC behavior, which will be reported in the next section. Indeed, the charge carrier transport along this material may be regarded as the composition of intra-GNP and inter-GNP contributions. The first refers to the conduction within individual graphene flakes. It can be approximated by Equation (1). Instead, the inter-GNP contribution considers the charge transport between two adjacent flakes. It involves two mechanisms: (i) a conduction current due to direct physical contact between the GNPs and (ii) tunneling or hopping currents activated when the GNPs are sufficiently close to each other (typically within ~0.34 nm, i.e., the Van der Waals distance). From an electrical perspective, these two effects can be modeled by two parallel resistances, which are then placed in series with the intrinsic intra-flake resistance described by Equation (1). The first is the contact resistance that can be modeled as [42]:(2)$$R_{cont}\left(T\right)=\frac{R_{0}+R_{P}(T)}{M\left(T\right)},$$
where the parasitic resistance $R_{P}(T)$ is strictly related to the quality of the contact and therefore increases with *T.* The tunneling and hopping resistance can instead be evaluated by using the Variable Range Hopping model [47]:(3)$$R_{tun,hop}\left(T\right)=\frac{K}{T^{1/4}},$$

*K* being a constant depending on the geometry.

By analyzing the resistance models (1)–(3), it is clear that the presence of counteracting phenomena can, in principle, lead to any sign in the Temperature Coefficient of the Resistance, TCR, that is defined as:(4)$$TCR\left(T\right)=\frac{1}{R(T)}\frac{dR(T)}{dT}.$$

Note that the TCR is often used to describe the temperature-sensitive properties, often defined with the difference approximation of (4), that is:(5)$$TCR\left(T\right)=\frac{∆R(T)}{R(T_{0})∆T}=\frac{1}{R\left(T_{0}\right)}\frac{R\left(T\right)-R\left(T_{0}\right)}{{T-T}_{0}},$$
being $T_{0}\;$ the reference temperature.

### 2.3. Experimental Setup for the Characterization of the Sensors

This paragraph describes the measurement system and methodologies employed for the thermo-electrical characterization of the GNP-based temperature sensor. A custom test fixture was used throughout the measurement campaign, enabling the stable attachment of the nanosensor electrodes and establishing consistent measurement points. This setup ensured system stability during testing, enhancing the results’ reliability. The test fixture was also designed to incorporate a four-terminal configuration; in Figure 3, A1 and A2 are the amperometric terminals, which are larger to reduce contact resistance, while V1 and V2 are the volumetric ones, which are thinner, to allow for precise identification of measurement points. The four-wire method inherently excludes contact resistances from the electrical circuit, so calibration is unnecessary to compensate for the contact resistances. However, parasitic contact resistances were preliminarily estimated to comprehensively characterize the system, as the difference in the resistance measured with both 2-wire and 4-wire techniques.

The linearity of the relationship between the applied current and the voltage drop across the GNP nanostrip is verified through a preliminary V-I characterization carried out in the current range of interest. This step allows the detection of possible non-linear issues during subsequent thermal-electric tests. The chosen measurement method for this characterization was the ammeter-voltmeter method, implemented using a SIGLENT SPS5041X DC (SIGILENT Technologies, Cochran Rd. Solon, OH, USA) power supply and two Agilent 34401A multimeters (Agilent Technologies, Santa Clara, CA, USA), as depicted in Figure 4. The DC power supply was used to set different current levels for the strip, while one multimeter, acting as an ammeter, provided accurate current readings, and the second multimeter served as a voltmeter to capture voltage values. Each device was remotely controlled via a GPIB-488 interface. A downstream voltmeter configuration was applied to minimize parasitic effects and maximize accuracy in the ammeter-voltmeter method. This configuration was crucial given the nanostrip’s nominal resistance, which was the same magnitude as the multimeter’s shunt resistance used for the current measurement. In an upstream configuration, the voltmeter would capture the nanostrip voltage and the ammeter’s voltage drop. The downstream setup, however, allowed the voltmeter to directly measure the total voltage drop across the strip’s terminals. The final voltage and current values were determined as the mean of multiple measurements for each set of current levels to obtain the GNP strip’s V-I characteristic.

The tests measured the sensor resistance at different temperature set points for the electro-thermal characterization. Multiple GNP strips were simultaneously placed in a programmable climatic chamber, and resistance measurements were taken at each temperature level for each strip. An Agilent 34401A multimeter, remotely controlled via the GPIB-488 interface, was employed to measure the resistance of each sensor using a four-wire measurement technique, minimizing the impact of contact resistances. The multimeter was set to its minimum range and highest resolution to achieve maximum accuracy. The environmental conditions during the tests were controlled using the ACS DY110 climatic chamber (Angelantoni, Massa Martana, Italy), which enabled precise adjustments of temperature and humidity levels. The setup is described in Figure 5.

For a more rigorous analysis, the GNP temperature sensor’s sensitivity to humidity was also characterized, as humidity fluctuations could influence resistance comparable to temperature variations, potentially confounding the results. Consequently, humidity-controlled measurements were also performed. The chamber’s thermocouple served as the temperature reference, and the chamber’s humidity sensor provided humidity measurements. The testing protocol consisted of a first phase where the temperature was varied without imposing a given humidity level. In the second phase, the thermal-resistive characterization was carried out by imposing fixed humidity values at given set points.

In the applied thermal cycle for the chosen temperature range, each temperature was reached first during the heating phase and then during the cooling phase to assess potential hysteresis. Once steady-state conditions were achieved at each temperature and humidity level, multiple measurements were taken, and the final values of resistance, temperature, and humidity were calculated as the mean of these measurements. A Fluke Hydra data logger was used to monitor the temperature with a K-type thermocouple placed on the top of each nanosensor in the climatic chamber to ensure steady-state conditions for each temperature. Measurements were taken only when the difference between the climatic chamber thermocouple and data logger readings was less than a predefined threshold, typically requiring around 20 min per level.

### 2.4. Sensor Model and Uncertainty Analysis

In this work, based on the considerations given in Section 2.2 and on the experimental results that will be presented in Section 3, the sensor’s calibration relation between temperature and resistance $\;T\left(R\right)\;$ is modeled as a polynomial function:(6)$$T\left(R\right)=\alpha_{0}+\alpha_{1}R+\alpha_{2}R^{2}+\cdots +\alpha_{p}R^{p},$$
where $\alpha =\left[\alpha_{0},\alpha_{1},\ldots ,\alpha_{p}\right]$ is the model parameter vector that is identified based on the experimental results, by means of the Modified Generalized Distance Regression (MGDR) algorithm [46]. The details of the calibration are discussed in Section 3.

Here, the model used for estimating the uncertainty is presented. The temperature *T* (dependent variable) and the resistance *R* (independent variable) are affected by measurement uncertainty. Consequently, the following equation can be written(7)$$T_{i}=\alpha_{0}+\alpha_{1}\cdot \left(R_{i}+\rho_{i}\right)+\alpha_{2}\cdot {\left(R_{i}+\rho_{i}\right)}^{2}+\cdots +\alpha_{p}\cdot {\left(R_{i}+\rho_{i}\right)}^{p}\;\xi_{i},\;\mathrm{for}\;i=1,\ldots ,n$$
where *n* is the number of the measured values of the resistance, *p* is the polynomial order (p+1<n), $\rho_{i}$ and $\xi_{i}$ are the (uncorrelated) measurement errors in *R* and *T*, respectively. The above-mentioned measurement errors belong to a Normal distribution with a mean equal to zero and a variance equal to $u_{R_{i}}^{2}$ and $u_{T_{i}}^{2}$ respectively:(8)$$\rho_{i}\in N\left(0,u_{r_{i}}^{2}\right)\mathrm{and}\;\xi_{i}\in N\left(0,u_{T_{i}}^{2}\right),\;\mathrm{for}\;i=1,\ldots ,n,$$
where $u_{R_{i}}$ and $u_{T_{i}}$ are the standard uncertainties of the mean values of *m* repeated acquisitions (in the present case *m* = 12).

The GDR technique consists of the evaluation of the vector α in order to minimize(9)$$\sum_{i=1}^{n}\left(\frac{{\left|{\hat{T}}_{i}-T_{i}\right|}^{2}}{u_{T_{i}}^{2}}+\frac{{\left|{\hat{R}}_{i}-R_{i}\right|}^{2}}{u_{R_{i}}^{2}}\right)=minimum,$$
where $\left({\hat{R}}_{i},{\hat{T}}_{i}\right)$ are the predicted points corresponding to the measured ones $\left(R_{i},T_{i}\right)$.

If correlation between uncertainty is present, (9) can be generalized as follows:(10)$$\left(\hat{T}-T\right)\cdot \Psi_{T}^{-1}\cdot {\left(\hat{T}-T\right)}^{T}+\left(\hat{R}-R\right)\cdot \Psi_{R}^{-1}\cdot {\left(\hat{R}-R\right)}^{T}=minimum$$
where $\hat{T}$, *T*, $\hat{R}$ and *R* are vectors of dimension $\left(1\times n\right)$, $\Psi_{T}$ and $\Psi_{R}$ are invertible matrixes of dimensions $\left(n\times n\right)$ and the apex *T* denotes the transposed vector, remembering that, from Equation (1), the predicted value ${\hat{T}}_{i}$ can be expressed as follows:(11)$${\hat{T}}_{i}={\hat{\alpha }}_{0}+{\hat{\alpha }}_{1}\cdot {\hat{R}}_{i}+\cdots +{\hat{\alpha }}_{p}\cdot {\hat{R}}_{p}^{2}\;\;\;\mathrm{for}\;i=1,\ldots ,n$$
where $\hat{\alpha }=\left[{\hat{\alpha }}_{0},{\hat{\alpha }}_{1},\ldots ,{\hat{\alpha }}_{p}\right]$ are the estimated values of the model parameters.

The polynomial order *p* can be a priori chosen by introducing the additional model uncertainty variance covariance matrix $\Psi_{mod}$:(12)$$\Psi_{mod}=u_{mod}^{2}\cdot I_{n}$$
where $I_{n}$ is the identity matrix of dimensions (*n* × *n*). Therefore,(13)$$\Psi_{R}^{*}=\Psi_{R}+u_{mod}^{2}\cdot I_{n}$$
and the $u_{mod}$ value is obtained through the compatibility index between predicted and measured values, assuring that [48](14)$$I_{T_{i}}=\frac{\left|{\hat{T}}_{i}-T_{i}\right|}{z_{i}\cdot \sqrt{u_{{\hat{T}}_{i}}^{2}+u_{T_{i}}^{2}}}\leq 1\;\;\mathrm{for}\;i\;=\;1,\ldots ,n,$$
where $I_{T_{i}}$ is the compatibility index on the temperatures and *z_i_* is the coverage factor for a confidence level of 95.45%. In the present case, since a set of 12 measured values was obtained for each investigated resistance level, a value of *z_i_* = 2.201 was considered for i=1,…,n by a Student’s t distribution. A more detailed description of the algorithm is in [48]. In this paper, the calibration curves were obtained through a polynomial degree *p* = 1.

## 3. Results and Discussion

### 3.1. Preliminary Characterization and Calibration

In this paragraph, the results of the preliminary characterization and calibration are presented. The following current values were applied to power the sensor to determine the V-I characteristics of the GNPs strips: [0.01, 0.02, 0.05, 0.07, 0.1, 0.2, 0.5, 0.7, 1] A. For each current set point, 10 voltage and current measurements were acquired in an ammeter-voltmeter configuration. The final V-I curve was generated by averaging all the measurements for each current level, showing a linear relationship for all samples. Figure 6 shows, as an example, the result obtained with G−PREG 50/50.

Finally, the contact resistances for each strip were estimated by calculating the difference between the average of 10 resistance values measured using a 2-wire technique and 10 values measured using a 4-wire (4W) technique. The results for all the materials are provided in Table 2 and demonstrate that a calibration of the measurement system is essential for 2-wire (2W) configurations, as the contact resistance significantly affects the measurement accuracy.

### 3.2. Characterization of the Sensing Elements and Identification of the Resistance Model

The electrical resistance of the three GNP strip formulations listed in Table 1 has been measured following the procedure detailed in paragraph 2.3. In particular, samples for each material have been tested by cycling the temperature from −40 °C to 60 °C, with a 10 °C step. The results of this characterization are plotted in Figure 7 for a sample of each of the three formulations and refer to the rising and falling branches of the temperature cycle. As expected, the minimum resistance value (in the range 1.1–1.4 Ω) is exhibited by the material with the highest percentage of GNPs (G−PREG 95/5). Increasing the percentage of insulating binder (G−PREG 70/30) leads obviously to an increase in the resistance (in the range of 2.1–2.5 Ω). In addition, as already pointed out, the boron nitride is employed in the G−PREG 50/50 to increase electrical resistivity without significantly reducing thermal conductivity. This material shows the highest resistance value (3.3–3.8 Ω).

Note that 10 measurements of resistance and temperature were collected and averaged for each temperature level to obtain the results. Moving from one temperature level to the next, it was required to reach steady-state conditions, which were confirmed by ensuring that the temperature difference between the chamber’s thermocouple and the data logger’s thermocouples was less than 0.5 °C. A medium stabilization time of 20 min was necessary at each temperature level.

A study on the repeatability of resistance measurements was conducted by evaluating the Type B uncertainty over 10 measurements at different temperature levels with no humidity control. This assessment was performed for each sample and formulation during the rising and falling cycles. The nanomaterial demonstrated a strong response in terms of repeatability across all conditions. For example, Table 3 presents the evaluated Type A and Type B uncertainties for a G−PREG 50/50 sample during the heating phase at each temperature level. The results show that the Type B uncertainty, associated with repeatability, has a negligible impact compared to the Type A uncertainty, indicating high robustness and consistency in the measurements. Similar results are found for the other two materials.

A hysteresis characterization was also performed to assess its influence on the measured resistance values. The hysteresis effect was quantified as the absolute difference between the resistance values measured during the rising and falling phases at the same temperature. This difference was then expressed as a percentage relative to the mean resistance value obtained during the heating phase. For each sample and formulation, the maximum error value was considered. The results, presented in Table 4, indicate that the impact of hysteresis is negligible, being the errors below 1% in all cases except for one sample in the G−PREG 95/5 formulation.

Although all three materials exhibit similar behavior, G−PREG 50/50 was selected for further investigation of the temperature sensor properties due to its superior performance in terms of relative uncertainty (see Table 5). The combined standard uncertainty was determined for each formulation and sample at each temperature level during the heating and cooling phases under conditions without humidity control. This evaluation considered Type A and Type B uncertainty components, as reported in Table 3. The worst-case scenario, i.e., the highest observed uncertainty, was considered for each sensor strip. Table 5 reports the relative uncertainty with respect to the measured resistance values.

The strips also exhibit an NTC behavior, with the resistance decreasing as temperature rises. This phenomenon is due to the quantum effects pointed out in Section 2. Additionally, it can be observed that hysteresis is minimal, having no significant effect on the response. Finally, it is also evident that a linear model for the relation R(T) would satisfactorily fit the measurements,(15)$$R\left(T\right)=R(T_{0})(1+\alpha \left({T-T}_{0}\right)),$$
being $T_{0}$ the reference temperature, and α = $\;TCR\left(T\right)$ the Temperature Coefficient of the Resistance as defined in (5). The result is based on the past findings of the authors, see [34,35,36], and suggests that the envisioned sensor has a linear response in the investigated range, which is a very favorable property for a sensor.

The sensor model and validation are discussed in the following paragraph. Here, we conclude the analysis of the experimental results by investigating the impact of humidity variation on the behavior of the G−PREG 50/50 strips. Resistance measurements at different humidity and temperature values were performed on two samples of this material. Specifically, the humidity was tested in the range [10 °C, 60 °C] at the following Relative Humidity (RH) values: [20%, 40%, 60%, 70%, 80%, 90%]. The results presented in Figure 8 were obtained as the average of 20 measurements (10 during the temperature rise and 10 during the fall) for each temperature value corresponding to the specified humidity level. As in previous tests, measurements were only taken once steady-state conditions were achieved, defined here as a temperature difference of less than 0.5 °C between the chamber’s thermocouple and the data logger’s thermocouples. However, this test required a longer stabilization time (approximately 30 min) compared to the previous test due to adjustments in the chamber’s humidity control.

### 3.3. Sensor Model Identification and Validation, and Uncertainty Analysis

Given the results in the previous paragraph, a linear model is proposed to describe the calibration curve of the proposed temperature sensor; therefore, (6) reduces to:(16)$$T\left(R\right)=\alpha_{0}+\alpha_{1}R.$$

The linear regression model in Equation (16) described by only two parameters, is a particular case of the more general polynomial model presented in Equation (6). This choice is justified by the results of the experimental characterization of the sensors (see Figure 7), which can be conveniently fitted by a linear model, as shown later. However, the uncertainty introduced by this choice is taken into account when evaluating the model uncertainty, as indicated in Equation (13).

The model parameters have been identified by using the resistance values measured at the temperatures of [−40°, −20°, 0°, 20°, 40°, 60°], while the resistance values acquired at temperatures of [−30°, −10°, 10°, 30°, 50°] were used for the model validation. The results are given in Figure 9 and refer to two different samples of the G−PREG 50/50 strips. The measurement points used to identify the model are reported with red dots and the uncertainty bar associated with the measurement uncertainty. The solid line is the linear model (16), whereas the dashed lines represent the theoretical uncertainty computed as indicated in paragraph 2.4, with a confidence level of 95%.

As shown in Figure 9, the model has been successfully validated for both samples since the measured values of the dataset used for the validation all fall within the interval predicted by the uncertainty.

Finally, the influence of the humidity on this response has been evaluated by testing the resistance variation in the range [10 °C, 60 °C] at the following Relative Humidity (RH) values: [20%, 40%, 60%, 70%, 80%, 90%]. The results are reported as colored dots in Figure 10, with the sensor response (red solid line) and the uncertainty. These results correspond to the best and worst cases obtained when analyzing the G−PREG 50/50 strips. Specifically, the best case is given by sample 2 (Figure 10, right) since for all the values of RH, the response falls within the range predicted by the uncertainty. Instead, the worst case is given by sample 1 (Figure 10, left), for which the effect of the humidity change is included in the uncertainty only when considering a narrower RH range (from 40% to 60%).

The sensor’s performance is quantified in Table 6, which reports the estimated uncertainties associated with the measured resistance, *u*(*R*), to the measured temperature, *u*(*T*), and the predicted temperature, *u*(Tpred). The latter is the key performance indicator for the sensor, and the results shown in Table 6 suggest that this strip can be used as a temperature sensor in the studied interval, with an uncertainty of about 3 °C. This performance can be further improved if the strips of G-Paper 50/50 are used as sensors for temperature variations; that is, (16) becomes.

(17)$$∆T=\alpha_{1}\cdot ∆R,$$
where $\alpha_{1}=-229.829{\cdot \Omega }^{-1}$ for the analyed strip. In this case, the quantities involved in the calculations are almost fully correlated, and the uncertainties are given by:(18)$$u\left(∆R\right)=\sqrt{{u\left(R_{2}\right)}^{2}+{u\left(R_{1}\right)}^{2}-u\left(R_{1}\right)\cdot u\left(R_{2}\right)}=u\left(R\right)≅0.004$$(19)$$u\left(∆T\right)=\sqrt{{\left[u\left(\alpha_{1}\right)\cdot ∆R\right]}^{2}+{\left[\alpha_{1}\cdot u\left(∆R\right)\right]}^{2}}≅0.97$$
being $u\left(\alpha_{1}\right)=0.512{\cdot \Omega }^{-1}$. In this way the sensors based on strips of G-Paper (50/50) would be able to sense temperature differences of about 2 °C.

## 4. Conclusions

This work investigated the potential dual use of thin graphene coatings, which are primarily employed for thermal management or electromagnetic shielding, by adding a secondary temperature-sensing functionality based on the variation in electrical resistance with temperature. Commercial graphene-based thin films were analyzed, consisting of graphene nanoplatelets (GNPs) mixed with binders in different proportions: (i) 95% GNPs and 5% polyurethane (G−PREG 95/5); (ii) 70% GNPs and 30% epoxy (G−PREG 70/30); (iii) 47.5% GNPs, 47.5% boron nitride, and 5% polyurethane (G−PREG 50/50).

All these materials were tested across the range of –40 °C to +60 °C, demonstrating a stable, monotonic, and linear Negative Temperature Coefficient (NTC) behavior, with a temperature coefficient of the resistance of the order of $-1.5\cdot 10^{-3}$ ${{}^{\circ}C}^{-1}$. A linear temperature–resistance model was developed and validated, yielding temperature prediction uncertainties in the range of ±2 to ±3 °C.

The effects of humidity were also analyzed: within the relative humidity (RH) range of 40% to 60%, the influence on resistance remained within the model’s uncertainty bounds, suggesting that the sensor is robust against moderate humidity variations. Further investigations would be needed to assess performance under wider humidity ranges.

In summary, the key findings of this work are

-The feasibility of temperature sensing using commercially available GNP-based coatings has been demonstrated.-A linear sensor model with quantified uncertainty has been proposed and validated.-The impact of humidity on the sensor’s performance has been quantified.

While the proposed GNP-based sensors are not intended to replace high-precision devices such as thermocouples or RTDs, they offer a cost-effective and scalable alternative for large-area monitoring, integration into composite materials, and retrofitting of industrial surfaces.

Future work will focus on characterizing additional performance metrics, such as response time, signal-to-noise ratio, and long-term stability, as well as expanding the range of operating conditions and application scenarios.

## Figures and Tables

**Figure 1 sensors-25-03227-f001:**
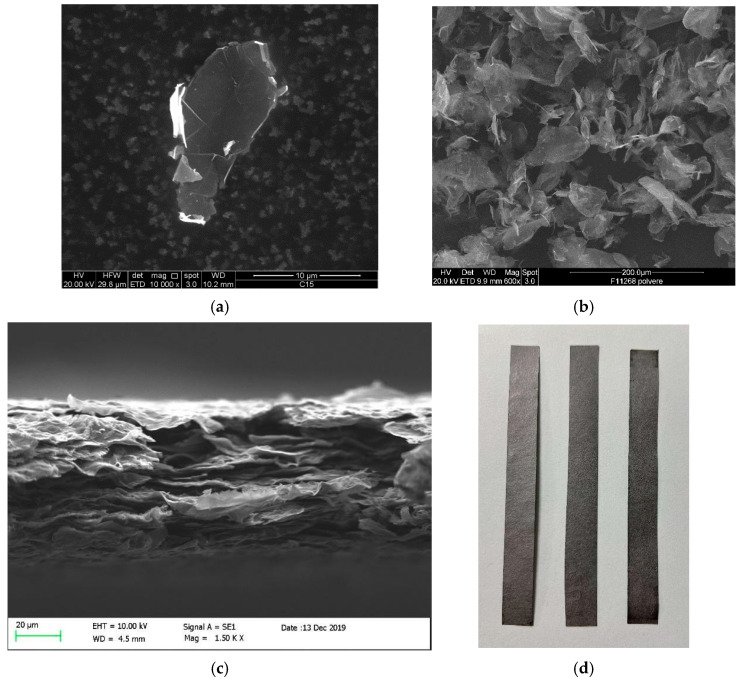
Scanning Electron Microscope characterization of the industrial graphene: single graphene nanoplatelet (**a**); random structure of the GNP agglomerate (**b**); lateral side of the resulting nano-composite material (**c**); macroscopic strips (**d**).

**Figure 2 sensors-25-03227-f002:**
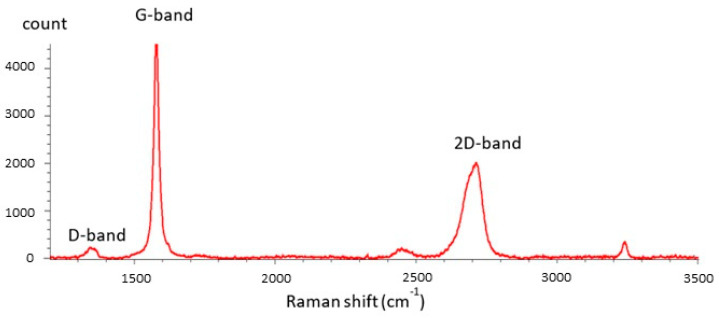
Raman spectroscopy characterization of the GNPs used in this work. The low intensity and broadness of the D-band, coupled with the high intensity and sharpness of the G-band indicates low concentration of defects in the surface of the material.

**Figure 3 sensors-25-03227-f003:**
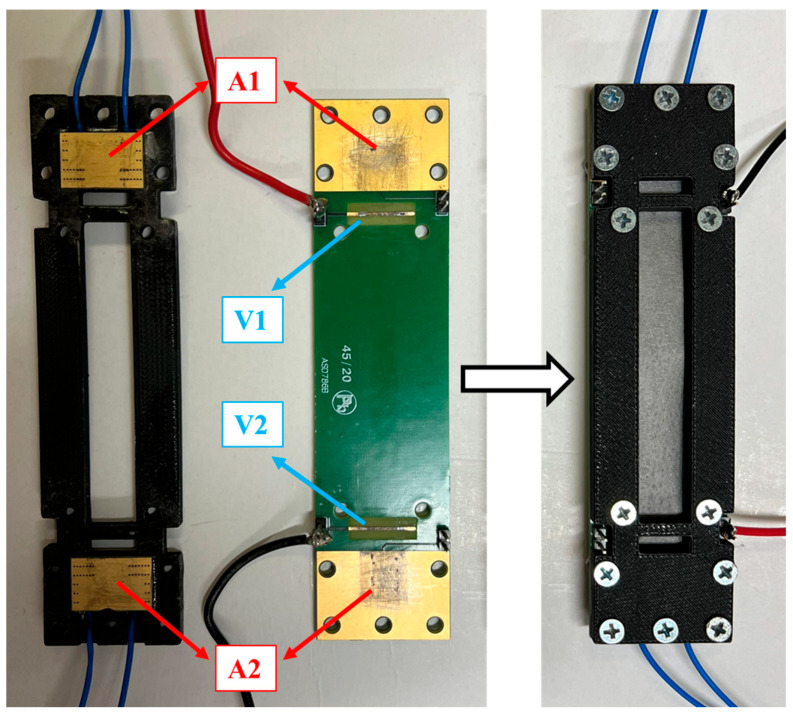
Test fixture designed to perform the four-probe measurement. Fixture open (left side) and with the sample inserted (right side).

**Figure 4 sensors-25-03227-f004:**
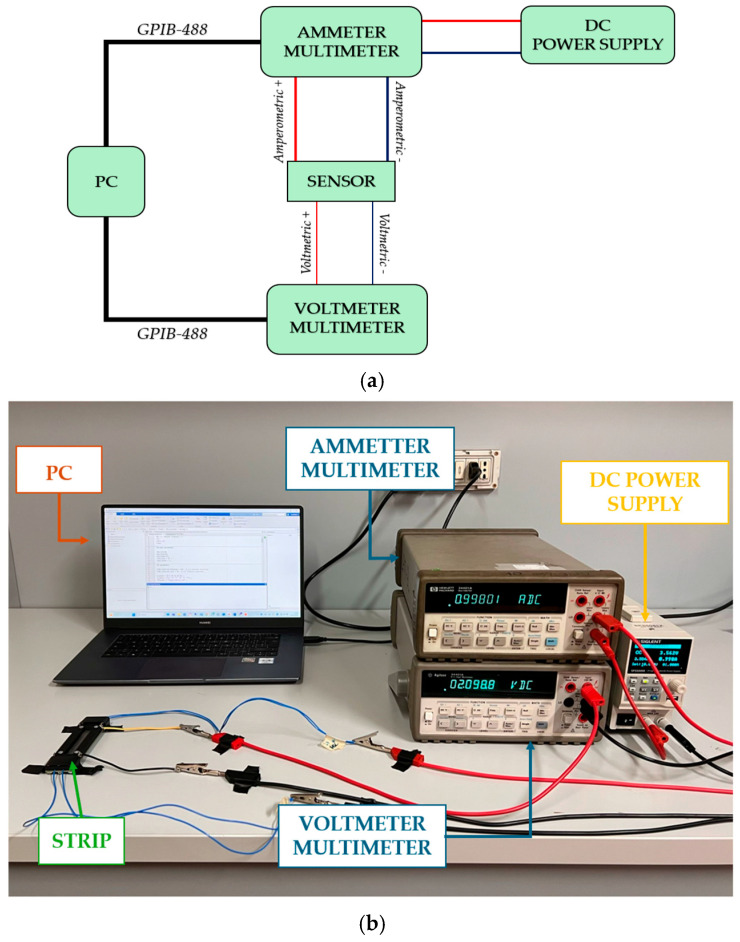
Schematic (**a**) and picture (**b**) of the setup for the preliminary electrical characterization.

**Figure 5 sensors-25-03227-f005:**
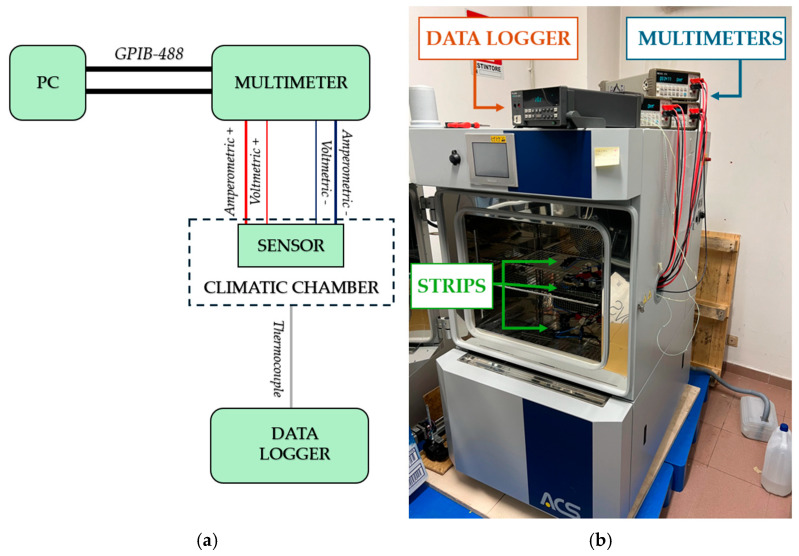
Schematic (**a**) and picture (**b**) of the setup used for the electro-thermal characterization.

**Figure 6 sensors-25-03227-f006:**
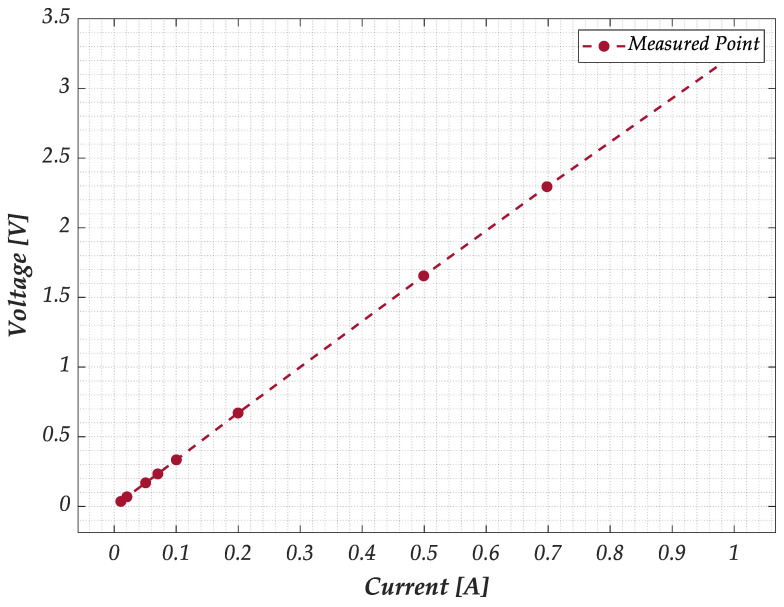
Preliminary characterization of the GNP strips: measured V-I characteristic for a sample of G−PREG 50/50 material.

**Figure 7 sensors-25-03227-f007:**
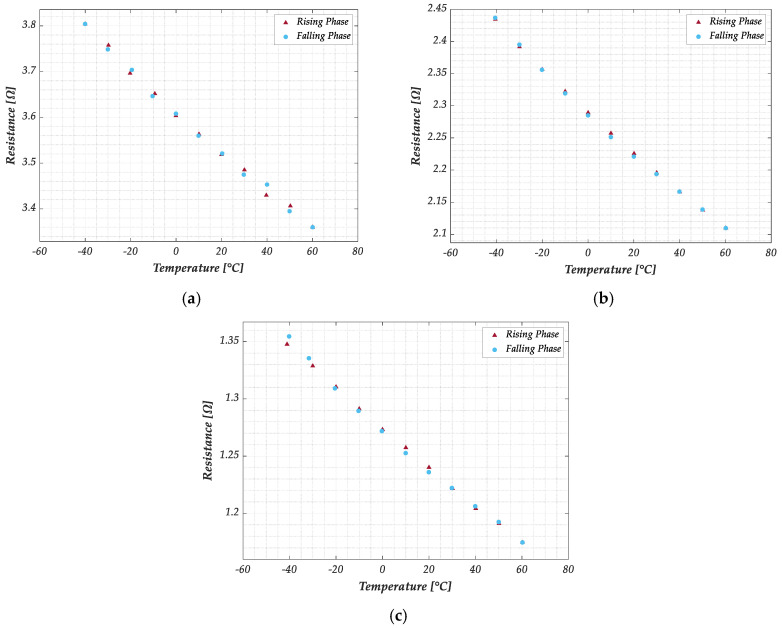
Measured electrical resistance of the graphene samples with temperature cycling in the range −40 °C to 60 °C: (**a**) G−PREG 50/50; (**b**) G−PREG 70/30; (**c**) G−PREG 95/5.

**Figure 8 sensors-25-03227-f008:**
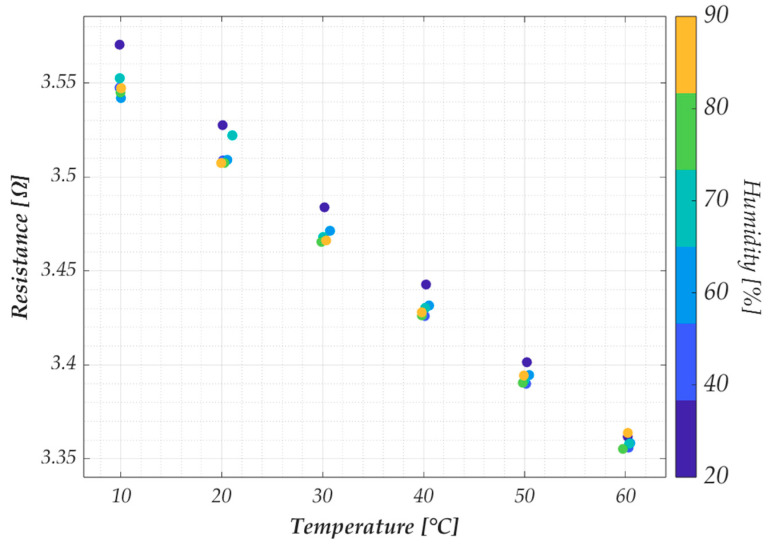
Measured electrical resistance for the G−PREG 50/50 in the temperature range from 10 °C to 60 °C, for varying values of the relative humidity.

**Figure 9 sensors-25-03227-f009:**
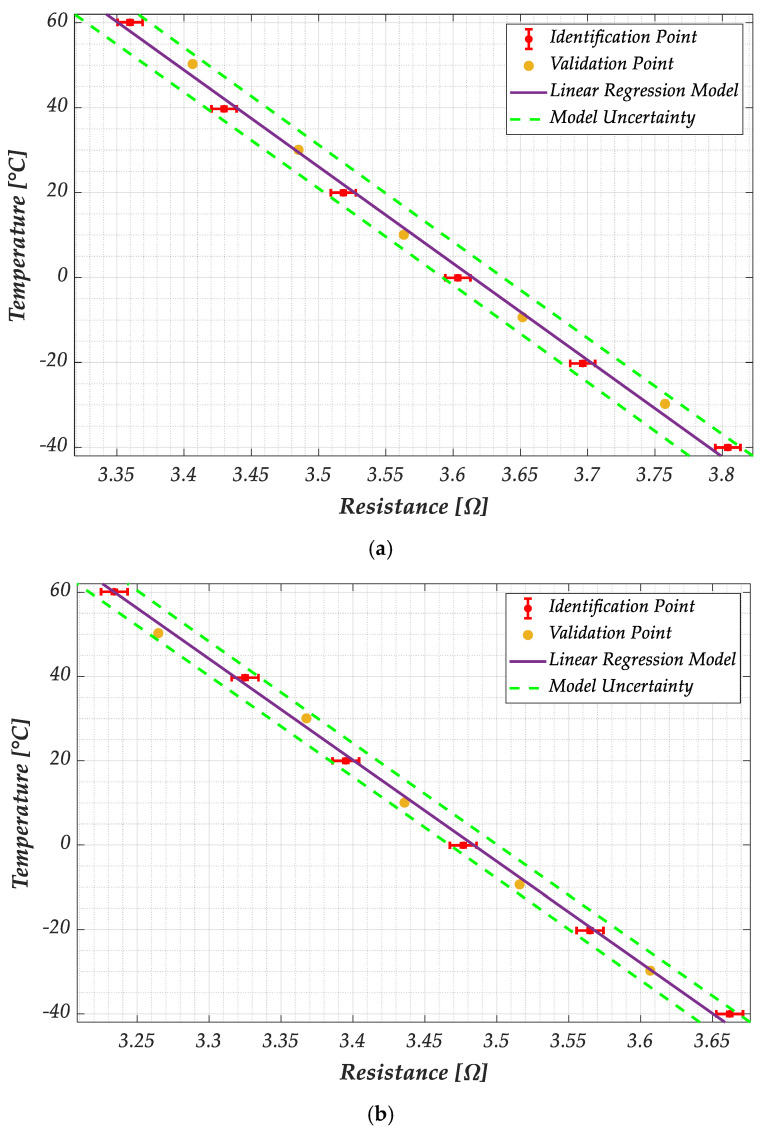
Plots of the temperature versus the electrical resistance according to the linear model in Equation (16), for the two considered samples of the G−PREG 50/50 films: (**a**) sample 1, and (**b**) sample 2. The red dots indicate the values adopted for identifying the model, whereas yellow dots are the values used to validate it. The red bars represent the uncertainty on the measurements, and the dashed green lines describe the estimated model uncertainty.

**Figure 10 sensors-25-03227-f010:**
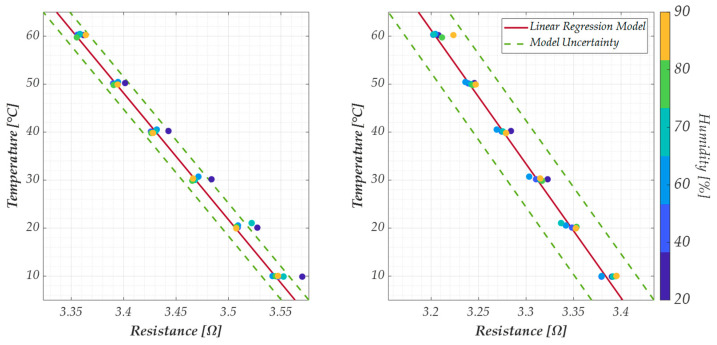
Plots of the temperature versus the electrical resistance according to the linear model in Equation (16), for the samples 1 (**left**) and 2 (**right**) of the G−PREG 50/50 films, for different values of relative humidity from 20% to 90%. Colored dots are the measured values, whereas the solid red line is the sensor response and the green dashed lines represent the model uncertainty.

**Table 1 sensors-25-03227-t001:** Characteristics of the graphene strips analyzed here, produced by Nanesa [21].

Material	%GNPs	Binder	Thickness (µm)	Length (cm)	Width (cm)
G-PREG (95/5)	95	Polyurethane 5%	94	10	1
G-PREG (70/30)	70	Epoxy 30%	85	10	1
G-PREG (50/50)	47.5	Boron nitride 47.5%, Polyurethane 5%	100	10	1

**Table 2 sensors-25-03227-t002:** Resistance values were measured through 2W and 4W techniques, and estimated contact resistance for all tested materials.

Material	# STRIP	Resistance 2W [Ω]	Resistance 4W [Ω]	Contact Resistance [Ω]	Contact Resistance/ Resistance 4W [%]
G-PREG (95/5)	1	1.62	1.20	0.42	35.00
	2	1.73	1.23	0.50	40.65
	3	1.69	1.21	0.48	39.66
G-PREG (70/30)	1	2.89	2.26	0.63	27.87
	2	2.61	2.04	0.57	27.94
	3	2.85	2.22	0.63	28.37
G-PREG (50/50)	1	4.23	3.43	0.80	23.32
	2	4.40	3.56	0.84	23.59
	3	4.09	3.35	0.74	22.08

**Table 3 sensors-25-03227-t003:** Type A and type B uncertainties are estimated for a sample of the G−PREG 50/50 strip under rising conditions.

*R*	*u_B_* (*R*)	*u_A_* (*R*)	*T*	*u_B_* (*T*)	*u_A_* (*T*)
(Ω)	(Ω)	(Ω)	(°C)	(°C)	(°C)
3.804	0.000	0.004	−40.030	0.014	0.230
3.758	0.001	0.004	−29.760	0.029	0.230
3.704	0.001	0.004	−19.480	0.053	0.230
3.652	0.000	0.004	−9.350	0.055	0.230
3.608	0.000	0.004	−0.060	0.021	0.230
3.563	0.000	0.004	10.070	0.014	0.230
3.521	0.000	0.004	20.280	0.044	0.230
3.485	0.000	0.004	30.070	0.040	0.230
3.453	0.000	0.004	40.030	0.028	0.230
3.406	0.000	0.004	50.290	0.026	0.230
3.360	0.000	0.004	60.100	0.000	0.230

**Table 4 sensors-25-03227-t004:** Maximum hysteresis error is evaluated for each strip and each formulation.

Material	# STRIP	Max Hysteresis Error [%]
G-PREG (95/5)	1	0.69
	2	1.17
	3	0.51
G-PREG (70/30)	1	0.24
	2	0.57
	3	0.26
G-PREG (50/50)	1	0.52
	2	0.68
	3	0.53

**Table 5 sensors-25-03227-t005:** Relative uncertainty (%) for each material.

Material	Relative Uncertainty [%]
G-PREG (50/50)	0.13
G-PREG (70/30)	0.21
G-PREG (95/5)	0.36

**Table 6 sensors-25-03227-t006:** Estimated composite and model uncertainties of the G−PREG 50/50 films.

*R*	*u* (*R*)	*T*	*u* (*T*)	Rpred	Tpred	*u* (Tpred)
(Ω)	(Ω)	(°C)	(°C)	(Ω)	(°C)	(°C)
3.899	0.004	−40.03	0.230	3.900	−42.19	2.923
3.794	0.004	−20.24	0.230	3.790	−21.16	2.922
3.683	0.004	−0.06	0.230	3.680	0.94	2.922
3.583	0.004	20.28	0.230	3.580	21.01	2.922
3.485	0.004	39.71	0.230	3.480	40.57	2.922
3.399	0.004	60.10	0.230	3.400	57.74	2.923

## Data Availability

The data presented in this study are available on request from the corresponding author. The data are not publicly available due to their use in ongoing projects.

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
