# Peer review of "Industrial-Grade Graphene Films as Distributed Temperature Sensors"

_sensors, 2025, doi:10.3390/s25103227_

Round 1
Reviewer 1 Report
Comments and Suggestions for Authors
This study investigated the possibility of using thin strips made from graphene nano platelets as low-cost distributed temperature sensors. The experimental characterization confirmed that the electrical resistance of the materials exhibits a negative temperature coefficient behaviour, with a linear dependence on temperature within the investigated temperature range (-40°C,+60°C).
The following suggestions are provided for revision:
(1) The abstract needs to include more specific results from this study.
(2) The SEM image in Figure 1 was taken in 2019 - please confirm it matches the material used in this study.
(3) For sample G-PREG (50/50) containing 47.5% boron nitride, please clarify its effect on sample properties.
Author Response
REV: This study investigated the possibility of using thin strips made from graphene nano platelets as low-cost distributed temperature sensors. The experimental characterization confirmed that the electrical resistance of the materials exhibits a negative temperature coefficient behaviour, with a linear dependence on temperature within the investigated temperature range (-40°C,+60°C). The following suggestions are provided for revision:
AU: We would like to thank the reviewer for comments and suggestions that allowed us to improve the work.
REV, query 1: The abstract needs to include more specific results from this study.
AU: The abstract has been completely revised. In the current version it contains a synthesis of the main findings, as suggested by the reviewer.
REV, query 2. The SEM image in Figure 1 was taken in 2019 - please confirm it matches the material used in this study.
AU: The SEM micrograph presented in Fig.1, although acquired in 2019, is still representative of the structure and high degree of alignment of the GNPs constituting the strips used in this work.
REV, query 3: For sample G-PREG (50/50) containing 47.5% boron nitride, please clarify its effect on sample properties.
AU: In the formulation of these samples, boron nitride is employed as a tuning agent for electrical resistance. Specifically, it is used to increase electrical resistivity without significantly reducing thermal conductivity.
This clarification has been added to the paper in this sentence before Figure 1:
“The third material is made by using a hybrid filler strategy, specifically by mixing GNPs and boron nitride (BN) at equal fractions (47.5%) together with a small percentage of polyurethane (5%). This formulation increases the strip's electrical resistance without lowering the thermal conductivity, leveraging BN's electrically insulating yet thermally conductive properties [30].”
Another sentence has been added right before Fig.7:
“In addition, as already pointed out, the boron nitride is employed in the G-PREG 50/50 to increase electrical resistivity without significantly reducing thermal conductivity. This material shows the highest resistance value (3.3-3.8 Ω).”
Reviewer 2 Report
Comments and Suggestions for Authors
The work is at a low scientific level, as the scientific contribution in the field of graphene temperature sensors is negligible. The authors used commercial material on which they measured the effect of temperature and humidity on electrical parameters in a climate chamber and applied basic statistical methods to the results. If the authors had avoided a lot of inaccuracies and vague statements, this could be considered a pretty good engineering paper, but not a scientific paper.
The introduction is very general and does not reflect the current state of the art in graphene-based temperature sensors. The authors should introduce what has been achieved in this area (what is the sensitivity of the sensors, resistance to moisture effects, long-term stability, etc.), or what is the current state of industrially produced graphene sensors. Furthermore, the introduction contains passages that should rather be in the experimental part and, on the contrary, should not contain descriptions of individual chapters. I consider the biggest mistake in the introduction to be that the authors do not state what the main contribution of their publication is in this area and how they distinguish themselves from other works of this type.
The article contains many vague, unclear, and sometimes false statements.
- The characterization of the material is absolutely insufficient. I consider it necessary to show the structure of the material - SEM of individual graphene flakes, or to demonstrate the individual materials (graphene, boron nitride) in the resulting material.
- References to literature are missing in important places, e.g. "The use of an industrial manufacturing process enables the production of materials that are more mechanically stable, have fewer defects, and offer good reproducibility compared to those produced in a laboratory setting". This statement must be supported by references.
- The false statement "then the thermal activation has no effect and the only mechanisms is the scattering: for these carbon materials, the electrical resistance is always a decreasing function of T. " This does not make sense, especially in the case of metallic SWCNTs. If the mechanism is only scattering, the resistance should increase with temperature.
- It is not stated for which material the volt-ampere characteristic is measured.
- It makes no sense to report the results of contact resistance measurements for G-PREG 70/30 material when the other measurements are performed for G-PREG 50/50 material.
- What does the term "no humidity control" mean? Was the humidity not adjusted during temperature measurement, or was it set to a constant value and a temperature test was performed?
- In the graphs of resistance versus temperature, it is not marked which points were measured for increasing temperature and which for decreasing temperature.
- I would assume that the final sample for testing will be selected based on some key sensor parameter and not on the highest resistance as the authors state. High resistance does not automatically mean better sensor parameters.
- Why are the temperature and resistance axes swapped in Figures 8 and 9? Without further explanation or a reason, it makes no sense.
- In the conclusions, the authors provide a TCR value for their sensor, which is incorrect. When I recalculated it, the TCR value comes out to be much smaller, even if I take into account deviations.
The English is at a fairly good level, but in some passages the authors could have expressed themselves more scientifically and not colloquially.
Author Response
REV: The work is at a low scientific level, as the scientific contribution in the field of graphene temperature sensors is negligible. The authors used commercial material on which they measured the effect of temperature and humidity on electrical parameters in a climate chamber and applied basic statistical methods to the results. If the authors had avoided a lot of inaccuracies and vague statements, this could be considered a pretty good engineering paper, but not a scientific paper.
AU: We would like to thank the reviewer for comments and suggestions that allowed us to improve the work.
REV, query 1: The introduction is very general and does not reflect the current state of the art in graphene-based temperature sensors. The authors should introduce what has been achieved in this area (what is the sensitivity of the sensors, resistance to moisture effects, long-term stability, etc.), or what is the current state of industrially produced graphene sensors. Furthermore, the introduction contains passages that should rather be in the experimental part and, on the contrary, should not contain descriptions of individual chapters. I consider the biggest mistake in the introduction to be that the authors do not state what the main contribution of their publication is in this area and how they distinguish themselves from other works of this type.
AU: The introduction has been completely revised, according to the reviewer’s suggestion. In particular, new references have been added and details about the performance of state of the art graphene sensors have been provided. In addition, it also contains a discussion about the use of industrial graphene, and a better clarification of the scope of the paper.
REV, query 2: The article contains many vague, unclear, and sometimes false statements. The characterization of the material is absolutely insufficient. I consider it necessary to show the structure of the material - SEM of individual graphene flakes, or to demonstrate the individual materials (graphene, boron nitride) in the resulting material.
AU: As requested by the reviewer, subsection 2.1 has been revised providing a detailed description of the fabrication process of the single GNPs and of the GNP films, along with their SEM characterization results. Two more SEM pictures have been added (see Fig.1a and 1b). In addition, a Raman spectroscopy analysis has been carried out and the resulting image has been added (see Fig.2), that allows the study of the quality of the material in terms of defects.
REV, query 3: References to literature are missing in important places, e.g. "The use of an industrial manufacturing process enables the production of materials that are more mechanically stable, have fewer defects, and offer good reproducibility compared to those produced in a laboratory setting". This statement must be supported by references.
AU: Following this suggestion, this statement has been now revised, and supported by a Raman characterization (Fig.2). The following sentence has been added to describe the characterization result (before Fig.2):
“The quality of the manufacturing process in terms of defects is checked by a Raman spectroscopy analysis, whose result is plotted in Fig.2. Indeed, the low intensity and broadness of the D-band associated with the high intensity and sharpness of the G-band indicate a low concentration of defects in the surface of the material, assessing the degree of structural disorder.”
REV, query 4: The false statement "then the thermal activation has no effect and the only mechanisms is the scattering: for these carbon materials, the electrical resistance is always a decreasing function of T. "This does not make sense, especially in the case of metallic SWCNTs. If the mechanism is only scattering, the resistance should increase with temperature.
AU: We would like to thank the reviewer for this comment, that helped us to correct a mistake. In the revised version, a clearer explanation of the physical mechanisms has been provided (see subsection 2.2), and in particular the following sentence has been added, that corrects the previous mistake:
“For instance, in metallic single-walled CNTs, the number of conduction channels M is almost independent of temperature [42]. Thus, the thermal activation has no effect, and the resistance increases with temperature, dominated only by the scattering.”
REV, query 5: It is not stated for which material the volt-ampere characteristic is measured.
AU: as requested by the reviewer, the material has been now specified in the caption of Fig.6 and in the text before such a figure.
REV, query 6: It makes no sense to report the results of contact resistance measurements for G-PREG 70/30 material when the other measurements are performed for G-PREG 50/50 material.
AU: following this suggestion, we have now added the contact resistance estimation for all the samples of all materials under investigation (see Table 2).
REV, query 7: What does the term "no humidity control" mean? Was the humidity not adjusted during temperature measurement, or was it set to a constant value and a temperature test was performed?
AU: we thank the review for this comment, that helped us to better clarify this point. The following sentence was added after Fig.4:
“The testing protocol consisted of a first phase where the temperature was varied without imposing a given humidity level. In the second phase, the thermal-resistive characterization was carried out by imposing fixed humidity values at given set points.”
REV, query 8: In the graphs of resistance versus temperature, it is not marked which points were measured for increasing temperature and which for decreasing temperature.
AU: the graphs (Fig.7) have been modified as suggested by the reviewer.
REV, query 9: I would assume that the final sample for testing will be selected based on some key sensor parameter and not on the highest resistance as the authors state. High resistance does not automatically mean better sensor parameters.
AU: the authors agree with this comment. Therefore, the choice of the material has been now better motivated, based on a better performance of the chosen material in terms of relative uncertainty. To this end, a new table has been added (Table 5) with the relative uncertainty and a comment about the choice has been added right before this Table:
“Although all three materials exhibit similar behavior, G-PREG 50/50 was selected for further investigation of the temperature sensor properties due to its superior performance in terms of relative uncertainty (see Table 5). The combined standard uncertainty was determined for each formulation and sample at each temperature level during the heating and cooling phases under conditions without humidity control. This evaluation considered Type A and Type B uncertainty components, as reported in Table 3. The worst-case scenario - i.e., the highest observed uncertainty - was considered for each sensor strip.”
REV, query 10: Why are the temperature and resistance axes swapped in Figures 8 and 9? Without further explanation or a reason, it makes no sense.
AU: following this comment, it has been now better clarified that the figures refer to the sensor model in Eq. (16), thus they show temperature as a function of the resistance.
REV, query 11: In the conclusions, the authors provide a TCR value for their sensor, which is incorrect. When I recalculated it, the TCR value comes out to be much smaller, even if I take into account deviations.
AU: the authors would like to thank the reviewer for this correction. Indeed, the first version of the manuscript reported wrong TRC values, that were associated to different materials. Now the values have been corrected (see the Abstract and the Conclusion sections)
Reviewer 3 Report
Comments and Suggestions for Authors
This study proposes graphene nanoplatelets (GNPs) as a low-cost distributed temperature sensing solution. By demonstrating the linear resistance-temperature relationship of graphene-based coatings, the work highlights their multifunctional potential for temperature monitoring applications. However, the structural integrity of these films formed by graphene-based coatings is constrained by challenges in achieving uniform continuity and precise thickness control, which negatively impacts the inherent mechanical and electrical properties of graphene as a two-dimensional material. Also, the integration of batch production feasibility, process complexity, and environmental impact is essential to assess the sensor's industrial viability. I suggest publishing it in Sensors after the changes suggested in the Comments section have been completed.

Author Response
REV: This study proposes graphene nanoplatelets (GNPs) as a low-cost distributed temperature sensing solution. By demonstrating the linear resistance-temperature relationship of graphene-based coatings, the work highlights their multifunctional potential for temperature monitoring applications. However, the structural integrity of these films formed by graphene-based coatings is constrained by challenges in achieving uniform continuity and precise thickness control, which negatively impacts the inherent mechanical and electrical properties of graphene as a two-dimensional material. Also, the integration of batch production feasibility, process complexity, and environmental impact is essential to assess the sensor's industrial viability. I suggest publishing it in Sensors after the changes suggested in the Comments section have been completed.
AU: We would like to thank the reviewer for comments and suggestions that allowed us to improve the work.
REV, query 1: The abstract of this paper does not clarify the innovation and highlights of this paper, which needs to be revised.
AU: The abstract has been completely revised. In the current version it contains a clearer synthesis of the main findings, as suggested by the reviewer.
REV, query 2: The introduction of this paper does not conform to the writing habits of scientific and technological papers and needs to be revised.
AU: The introduction has been completely revised, according to the reviewer’s suggestion. In particular, new references have been added and details about the performance of state of the art graphene sensors have been provided. In addition, it also contains a discussion about the use of industrial graphene, and a better clarification of the scope of the paper.
REV, query 3: The author proposed in the paper that” From the theoretical point of view, the mobility of charge carriers in carbon nanomaterials depends on the temperature following two counteracting mechanisms: thermal activation and scattering”. The author needs to provide a detailed explanation of the changing carrier mobility in graphene.
AU: Following the reviewer’s suggestion, a more detailed explanation of the physical mechanisms has been now provided (see subsection 2.2), with proper references to the literature.
REV, query 4: An SEM planar image and the results of Raman testing of graphene should be added.
AU: As requested by the reviewer, in subsection 2.1 two more SEM pictures have been added (see Fig.1a and 1b), to show the single GNP and the planar view of the GNP film. In addition, the results of a Raman spectroscopy analysis have been reported adding a new figure (see Fig.2), and a discussion has been provided in the text about these results.
REV, query 5: Additional sensor performance test data is needed. Such as sensor response time, repeatability, hysteresis, and SNR.
AU: As requested by the reviewer, additional data have been provided related to the sensor’s performance, by adding new tables and related comments. Specifically, Table 3 reports the results of a repeatability analysis over these sensors, commented in the text before the Table. Table 4 shows the maximum errors associated to the sensors’ hysteresis, that is indeed negligible, as highlighted in the comment before the Table. Table 5 presents the relative uncertainty for each analyzed material. The study carried out in this paper was only referred to steady-state conditions, but future work will address the study of the time evolution, to determine important parameters such as response time, long-term stability and so on, as pointed out in the Conclusion section.
REV, query 6: In Line 2 of Paragraph 3 of Page 7, "The industrial graphene composites analyzed in this paper are in the shape of strips 1 cm wide and 10 cm long with a thickness in the range of 85-100 μm. " The reason for choosing that size needs to be provided.
AU: to answer to this question, the following sentence has been added before Fig.1:
“The reason for using 1 cm x 10 cm GNP strips is related to the need to investigate macroscopic samples with an aspect ratio that simplifies the measurement of the electrical resistance, allowing the use of a simple test fixture (see subsection 2.3). Instead, the sheet thickness is only determined by the selection of standardized formulations provided by the manufacturer. “
REV, query 7: The physical drawings in Figures 3b and 4b should be labeled with the names of the corresponding instruments against Figures 3a and 4a.
AU: these figures (now 4b and 5b) have been revised as requested by the reviewer.
REV, query 8: In Line 6 of Paragraph 1 of Page 3,there is a reference to“GNPs are then dispersed in either acetone or an aqueous solution.”Please detail the reasons for dispersing GNP in acetone or aqueous solutions.
AU: to answer to this question, the following sentence has been added in the subsection 2.1
“GNPs are then dispersed in acetone or an aqueous solution to be subjected to sonication; a polymeric binder is included during this phase. The choice of solvent depends on the desired formulation: acetone is preferred for its fast evaporation rate and better compatibility with hydrophobic polymer matrices, while aqueous dispersions are more suitable when using hydrophilic additives, such as boron nitride and for environmentally friendly processing [30].
REV, query 9: In Line 6 of Paragraph 1 of Page 3, there is a reference to“To realize GNP sheets, the mixture is sprayed at a controlled pressure and the calendaring is used to compact the strips and optimize their thickness/alignment ratio.”Please describe in detail the operation of this step, indicating the exact value of the controlled pressure.
AU: to answer to this question, the following sentence has been added in the subsection 2.1
“To produce GNP sheets, the liquid formulation is deposited onto a release substrate using a controlled spray system operated by a pantograph. This setup allows for automatic deposition with controlled coating weight. Key process parameters—air and liquid pressures, nozzle-to-substrate distance, movement speed, and number of passes—are preset based on the specific product. Typically, the pressure values range from 0.9 to 1.5 bar, the nozzle-to-substrate distance from 10 to 20 cm, the movement speed from 60 to 120 cm/min, and the number of deposition cycles from 1 to 8. Following the spray deposition, the coating is compacted via cold calendering, applying a uniform pressure of up to 10 kN/m. The calendering process compacts the strips and optimizes their thickness-to-alignment ratio, improving electrical conductivity and film homogeneity [31].”
REV, query 10: Provide a specific process for the simplification of Eq. (6) to Eq. (16).
AU: as suggested by the reviewer, this choice has been now clarified by adding the following comment right after Eq. (16).
“The linear regression model in Eq. (16) described by only two parameters is a particular case of the more general polynomial model presented in Eq. (6). This choice is justified by the results of the experimental characterization of the sensors (see Fig.7), that can be conveniently fitted by a linear model, as shown later. However, the uncertainty introduced by this choice is taken into account when evaluating the model uncertainty, as indicated in Eq. (13).”
REV, query 11: Standardize the formatting of charts and graphs in this paper.
AU: all charts and graphs have been revised and standardized as requested by the reviewer.
REV, query 12: The conclusions section should concisely summarize the key findings and their significance in a concise manner. Please simplify the abstract in this paper.
AU: as suggested by the reviewer, the abstract has been simplified. In addition, the conclusion section has been revised to include a bullet list of the key findings.
Round 2
Reviewer 2 Report
Comments and Suggestions for Authors
- Incorrect units of thermal coefficient of resistance (TCR) are given in the abstract and conclusion.
- In the introduction it is not clear what "linearity 0.999" means. Should it be the value of the coefficient of determination? If so, it must be stated in the correct form.
- The unit "C°-1" occurs several times in the introduction, what does it mean? Is there supposed to be a superscript?
- There are statements in the description of the Raman spectrum (the size and width of the D-peak is related to defects in the structure) that you need to explain in detail if you have found them, or cite literature where they have clarified this.
- Table two gives the relative error (%). It is not clear what this error relates to. It would be useful to explain this. It would also be useful to write in the table the values of the resistances measured at 2W and 4W from which the contact resistance is determined.
- In the manuscript there are two tables labeled identically "Table 5" , this is confusing.
- When describing Figure 10, the authors state that the sensor response is shown as a blue solid line, but in the figure it is red. This is confusing.
- Figure 10 (left) shows a humidity value outside the tolerance interval other than the 20% RH. This should also be commented on.
Author Response
REV, query 1. Incorrect units of thermal coefficient of resistance (TCR) are given in the abstract and conclusion.
AU: The authors thank the reviewer for this remark. The TCR units have been now corrected both in the abstract and in the conclusion.
REV, query 2. In the introduction it is not clear what "linearity 0.999" means. Should it be the value of the coefficient of determination? If so, it must be stated in the correct form.
AU: following this suggestion, the introduction has been revised as follows:
“For instance, a graphene-polydimethylsiloxane composite doped with polyaniline has been used to realize a highly linear (R2=0.999, where R2 is the coefficient of determination, i.e. the square of the sample correlation coefficient)”
REV, query 3. The unit "C°-1" occurs several times in the introduction, what does it mean Is there supposed to be a superscript?
AU: following this suggestion, we have corrected the notation to make clear that -1 is a superscript.
REV, query 4. There are statements in the description of the Raman spectrum (the size and width of the D-peak is related to defects in the structure) that you need to explain in detail if you have found them, or cite literature where they have clarified this.
AU: following this suggestion, we have added two new references where this question is discussed and revised the following sentence in the subsection 2.1:
“Indeed, the low intensity and broadness of the D-band associated with the high intensity and sharpness of the G-band indicate a low concentration of defects in the surface of the material, assessing the degree of structural disorder, as shown in [34-35]”
REV, query 5. Table two gives the relative error (%). It is not clear what this error relates to. It would be useful to explain this. It would also be useful to write in the table the values of the resistances measured at 2W and 4W from which the contact resistance is determined.
AU: according to this request, Table 2 has been revised by adding the results coming from 2W and 4W measurements. In addition, it is now clarified that the last column is the ratio between the contact resistance and the 4W-resistance, expressed in %.
REV, query 6. In the manuscript there are two tables labeled identically "Table 5" , this is confusing.
AU: this mistake has been corrected
REV, query 7. When describing Figure 10, the authors state that the sensor response is shown as a blue solid line, but in the figure it is red. This is confusing.
AU: the caption of Fig.10 has been revised and the mistake has been corrected.
REV, query 8. Figure 10 (left) shows a humidity value outside the tolerance interval other than the 20% RH. This should also be commented on.
AU: We would like to thank the reviewer for this comment that allowed to better clarify the range of relative humidity where the humidity impact is accounted within the uncertainty. The following sentence has been added before Fig.10
“These results correspond to the best and worst cases obtained when analyzing the G-PREG 50/50 strips. Specifically, the best case is given by sample 2 (Fig.10, right) since for all the values of RH the response falls within the range predicted by the uncertainty. Instead, the worst case is given by sample 1 (Fig.10, left), for which the effect of the humidity change is included in the uncertainty only when considering a narrower RH range (from 40% to 60%).”
Consequently, the following sentence has been added to the abstract
“the errors introduced by relative humidity values in the range from 40% to 60% are included in the model’s uncertainty bounds”
and the following one to the conclusion:
“The effects of humidity were also analyzed: within the relative humidity (RH) range of 40% to 60%, the influence on resistance remained within the model’s uncertainty bounds, suggesting that the sensor is robust against moderate humidity variations.”
Reviewer 3 Report
Comments and Suggestions for Authors
The article can be accepted.
Author Response
REV: The article can be accepted.
AU: We would like to thank the reviewer for comments and suggestions that allowed us to improve the work.